# Estimation of the Two-Dimensional Direction of Arrival for Low-Elevation and Non-Low-Elevation Targets Based on Dilated Convolutional Networks

**Guoping Hu** [1], **Fangzheng Zhao** [1] and **Bingqi Liu** [2,*]

1 Air and Missile Defense College, Air Force Engineering University, Xi'an 710043, China
2 Graduate School, Air Force Engineering University, Xi'an 710043, China
* Correspondence: bingqi0828_liu@163.com

**Abstract:** This paper addresses the problem of the two-dimensional direction-of-arrival (2D DOA) estimation of low-elevation or non-low-elevation targets using L-shaped uniform and sparse arrays by analyzing the signal models' features and their mapping to 2D DOA. This paper proposes a 2D DOA estimation algorithm based on the dilated convolutional network model, which consists of two components: a dilated convolutional autoencoder and a dilated convolutional neural network. If there are targets at low elevation, the dilated convolutional autoencoder suppresses the multipath signal and outputs a new signal covariance matrix as the input of the dilated convolutional neural network to directly perform 2D DOA estimation in the absence of a low-elevation target. The algorithm employs 3D convolution to fully retain and extract features. The simulation experiments and the analysis of their results revealed that for both L-shaped uniform and L-shaped sparse arrays, the dilated convolutional autoencoder could effectively suppress the multipath signals without affecting the direct wave and non-low-elevation targets, whereas the dilated convolutional neural network could effectively achieve 2D DOA estimation with a matching rate and an effective ratio of pitch and azimuth angles close to 100% without the need for additional parameter matching. Under the condition of a low signal-to-noise ratio, the estimation accuracy of the proposed algorithm was significantly higher than that of the traditional DOA estimation.

**Keywords:** 2D DOA estimation; low-elevation-angle targets; L-shaped uniform array; L-shaped sparse array; dilated convolutional autoencoder; dilated convolutional neural network; 3D convolution

## 1. Introduction

Array signal processing, which has a wide range of applications in communications, remote sensing, detection, and radar, involves the use of sensor arrays to achieve signal parameter estimation, signal enhancement [1], etc. Accordingly, direction-of-arrival (DOA) estimation is an important branch of research. This involves estimating the direction of arrival of one or more signals in a region of space using theoretical or technical methods. One-dimensional (1D) DOA estimation is the estimation of the elevation angle of targets. Two-dimensional (2D) DOA estimation, as an extension of 1D DOA estimation, enables the estimation of both the elevation and the azimuth angles [2]. Two-dimensional DOA is of greater importance for spatial localization and is, therefore, one of the main focuses of current research in the field. Two-dimensional DOA estimation requires arrays to be arranged in a 2D plane, generally using L-shaped arrays, surface arrays, parallel arrays, or vector sensors [3,4]. Most 2D DOA estimation algorithms extend the 1D DOA estimation algorithm to a 2D spatial spectrum, such as the 2D multiple-signal classification (MUSIC) [5] algorithm and 2D estimating signal parameter via rotational invariance techniques (ESPRIT) algorithm. The former can produce asymptotic unbiased estimates with high estimation accuracy without the need for parameter matching, but this algorithm requires a 2D spatial spectrum

search and has a high computational demand. On the other hand, the latter does not require a spatial spectral search, and the elevation and azimuth angles can also be automatically matched, but the estimation accuracy of this method is low, especially when the signal-to-noise ratio (SNR) is low. Yin et al. proposed a DOA direction matrix method [6], where the elevation and azimuth angles could be directly obtained via eigendecomposition of the DOA direction matrix, with automatic parameter matching; however, this method is only applicable to specific arrays such as parallel linear arrays. To improve the estimation accuracy and spatial freedom, sparse arrays are often used in practice instead of uniform arrays [7]. In a study on the 2D DOA estimation of sparse arrays, Liu et al. proposed a 2D DOA estimation method based on singular value decomposition [8], taking advantage of the structural characteristics of T-shaped arrays and co-prime array arrays to obtain three signal subspaces without using virtual elements before using the signal subspaces to perform 2D DOA estimation. Wang et al. designed a generalized coprime parallel linear array instead of the traditional parallel uniform linear array, then improved the differential virtual array to obtain greater degrees of freedom, and finally simplified the 2D search to two 1D searches to reduce the number of operations [9]. However, the algorithm led to an increase in the influence of the mutual coupling between array elements, and the compression factor needed to be artificially chosen, restricting the performance of the algorithm. In addition, when the elevation angle of the target incident array is low, multipath effects can occur, which can result in the received signal including reflected waves that are coherent with the direct wave, thereby complicating 2D DOA estimation. For the 2D DOA estimation problem of low-elevation-angle targets, Ma et al. proposed a 2D DOA estimation algorithm based on the alternating direction method of multipliers [10], which transforms the 2D DOA estimation into two 1D DOA estimation problems and avoids the problem of the high computational demand caused by 2D joint estimation; however, the algorithm could only solve the 2D DOA estimation of a single target. Su et al. and Park et al. proposed 2D DOA estimation algorithms for coherent signals based on sparse reconstruction [11,12], which could be used for the decoherence of low-elevation targets; however, the algorithms had a complex arithmetic process. Liang et al. proposed a 2D DOA estimation algorithm for coherent sources based on Toeplitz matrix reconstruction [13], which could estimate the elevation and azimuth angles without loss of array aperture through a 1D search only; however, the algorithm was only applicable to uniform arrays. Molaei et al. proposed a k-medoids clustering signal separation method that could realize the 2D DOA estimation of multipath signals and effectively separate coherent and noncoherent signals [14]; however, the method was only applicable to rectangular arrays.

Usually, physical model algorithms suffer from limited applicability and complex computational processes, whereas data-driven deep-learning-based algorithms have greater applicability. Compared with traditional signal processing algorithms, deep-learning-like algorithms convert the DOA estimation problem into a high-dimensional nonlinear mapping relationship, i.e., realizing mapping between the covariance matrix of the received signal or other variables and the DOA, which provides a new way of thinking for the study of 2D DOA estimation methods. Marija et al. implemented the fast estimation of spatial single-target 2D DOA using a multilayer perceptron [15]; however, the artificial neural network (ANN) model needed to expand the signal covariance matrix into 1D data as input, thereby losing the spatial characteristics of the covariance matrix. Zhu proposed a 2D DOA estimation algorithm based on deep ensemble learning [16], using multiple convolutional neural networks to output the elevation and azimuth angles. This approach was not limited by the deployment method; however, there was a matching problem of elevation and azimuth angles.

To address the practical problems of the above algorithms, this paper proposes a 2D DOA estimation model based on the combination of a dilated convolutional autoencoder and a dilated convolutional neural network, whereby the former solves the coherence

problem of direct and reflected waves by suppressing the multipath signal, i.e., filtering out the reflected wave components of the signal covariance matrix, while the latter is used to implement 2D DOA estimation. Both the dilated convolutional autoencoder and the dilated convolutional neural network are convolved in three dimensions to fully extract the spatial features of the data; accordingly, the model is able to achieve the 2D DOA estimation of non-low-elevation targets and hybrid targets in L-shaped uniform and L-shaped sparse arrays without the need for parameter matching.

## 2. Signal Model

### 2.1. L-Shaped Array Signal Model

When the array arrangement is in one dimension, only 1D DOA estimation can be realized. If 2D DOA estimation is required for the source, i.e., elevation and azimuth, the array arrangement needs to be at least 2D. In this study, an L-shaped array was designed to perform 2D DOA estimation. When the L-shaped array consists of two mutually perpendicular uniform line arrays, its arrangement is as shown in Figure 1.

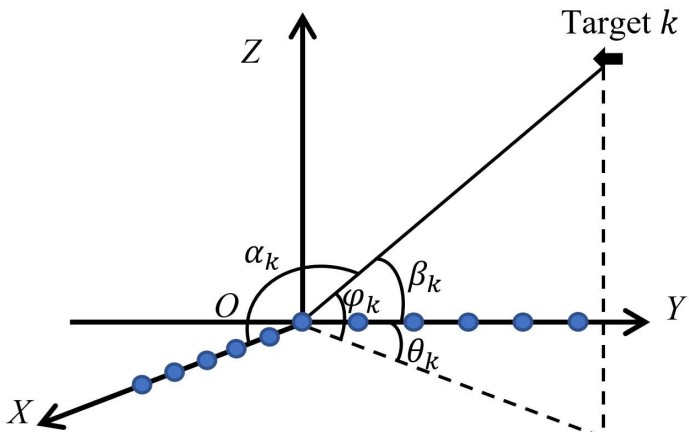

**Figure 1.** L-shaped array arrangement.

The array elements are uniformly arranged along the $X$ and $Y$ axes, with an array element spacing of $d$ and less than half a wavelength $\lambda$; and the number of $X$- and $Y$-axis array elements are $M$ and $N$, respectively, with overlapping array elements at the origin of the coordinate axis, so the total number of array elements is $(M + N - 1)$. Consider $K(K < M + N - 1)$ far-field uncorrelated narrowband signals incident to the L-shaped array in the directions $(\alpha_1, \beta_1), (\alpha_2, \beta_2) \ldots (\alpha_K, \beta_K)$ $(k = 1, 2, \ldots, K)$, where $\alpha_k$ and $\beta_k$ are the angles of the target to the $X$ and $Y$ axis, respectively, also known as the spatial phase factor; $\varphi_k$ and $\theta_k$ denote the elevation and azimuth angles of the target, respectively; and the correspondence between $\alpha_k$, $\beta_k$ and $\varphi_k$, $\theta_k$ is shown below:

$$cos\alpha_k = sin\varphi_k cos\theta_k, \tag{1}$$

$$cos\beta_k = sin\varphi_k sin\theta_k, \tag{2}$$

The received signals for a uniform line array along the $X$- and $Y$-axis directions are

$$x_1(t) = A_1 s(t) + n_1(t), \tag{3}$$

$$x_2(t) = A_2 s(t) + n_2(t), \tag{4}$$

where $s(t) = [s_1(t), s_2(t), \ldots, s_K(t)]^T$ denotes the signal vector; $n_1(t)$ and $n_2(t)$ denote gaussian white noise with noise power $\sigma^2$ and uncorrelated with the signal, respectively;

and $A_1$ and $A_2$ are the direction vectors of uniform line arrays in the $X$- and $Y$-axis directions, respectively.

$$A_1 = [a(\alpha_1), a(\alpha_2), \ldots, a(\alpha_K)], \tag{5}$$

$$A_2 = [a(\beta_1), a(\beta_2), \ldots, a(\beta_K)], \tag{6}$$

$$X: a(\alpha_k) = \left[1, e^{-\frac{j2\pi dcos\alpha_k}{\lambda}}, \ldots, e^{-\frac{j2\pi(M-1)dcos\alpha_k}{\lambda}}\right]^T, \tag{7}$$

$$Y: a(\beta_k) = \left[e^{-\frac{j2\pi dcos\beta_k}{\lambda}}, e^{-\frac{j2\pi 2dcos\beta_k}{\lambda}} \ldots, e^{-\frac{j2\pi(N-1)dcos\beta_k}{\lambda}}\right]^T, \tag{8}$$

Combining Equations (3) and (4) yields

$$x(t) = \left[x_1{}^H(t)x_2{}^H(t)\right]^T = B(\varphi, \theta)s(t) + n(t), \tag{9}$$

where $B(\varphi, \theta) = \left[A_1{}^H, A_2{}^H\right]^T$ and $n(t) = \left[n_1{}^H(t), n_2{}^H(t)\right]^T$, calculate the received signal covariance matrix according to Equation (9), i.e.,

$$R_x = E\left[x(t)x^H(t)\right] = B(\varphi, \theta)R_s B^H(\varphi, \theta) + \sigma^2 I_{M+N-1}, \tag{10}$$

where $R_s = E\left[s(t)s^H(t)\right]$ denotes the incident signal covariance matrix, and $I_{M+N-1}$ denotes the unit matrix of dimension $M + N - 1$. The eigendecomposition of the received signal covariance matrix $R_x$ can be divided into a signal subspace and a noise subspace,

$$R_x = U\Sigma U^H = U_s \Sigma_s U_s^H + U_n \Sigma_n U_n^H, \tag{11}$$

where $\Sigma$ denotes the diagonal matrix constructed from all the eigenvalues obtained from the eigen decomposition; $U$ denotes the eigenvector matrix; $\Sigma_s$ denotes the diagonal matrix constructed from the K largest eigenvalues in $\Sigma$ equal to the number of signals; $U_s$ denotes the eigenvector corresponding to the K largest eigenvalues, considered as the signal subspace; $\Sigma_n$ denotes the diagonal matrix constructed from the remaining $(M + N - 1 - K)$ eigenvalues; $U_n$ the eigenvectors corresponding to the remaining eigenvalues, which are regarded as the noise subspace. According to the theory of the MUSIC algorithm, the signal subspace and the noise subspace have orthogonal properties, and $U_n$ is orthogonal to $b(\varphi, \theta)$ column vector in $B(\varphi, \theta)$, and the spatial spectrum $P(\varphi, \theta)$ is calculated according to the 2D MUSIC algorithm, as follows

$$b(\varphi, \theta) = \left[a^H(\alpha), a^H(\beta)\right]^T, \tag{12}$$

$$P(\varphi, \theta) == \frac{1}{b^H(\varphi, \theta)U_n U_n^H b(\varphi, \theta)}, \tag{13}$$

The elevation and azimuth angles $(\varphi, \theta)$ can be obtained by searching for the peak points of the 2D spatial spectrum within the target airspace. Taking an L-shaped uniform array with the number of elements in the $X$ and $Y$ axis being 8 and 9, respectively, as an example, when three targets with elevation and azimuth angles of $(10°, 30°)$, $(20°, 10°)$, and $(40°, 20°)$ are incident on the array with SNR = 10 dB and snapshots = 100, the spatial spectrum and its top view were obtained after a 2D search, as shown in Figure 2.

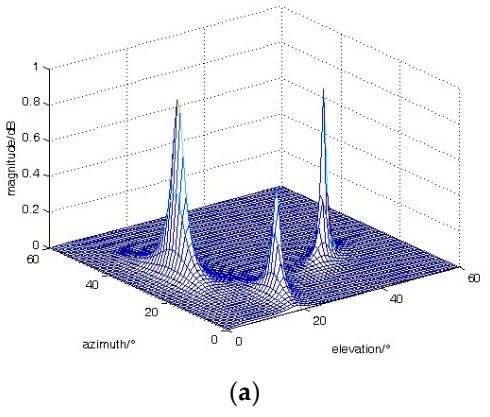

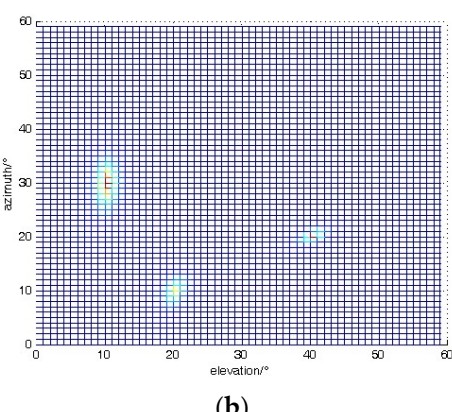

(**a**)  (**b**)

**Figure 2.** Spatial spectrum and top view of L-shaped uniform array. (**a**) Spatial spectrum of L-shaped uniform array. (**b**) Top view of L-shaped uniform array.

Figure 2 shows that the 2D MUSIC algorithm can effectively estimate elevation and azimuth with a high degree of accuracy. In practice, sparse arrays are often used instead of uniform line arrays to reduce the effect of the mutual coupling between array elements on the accuracy of DOA estimation and to increase the number of measurable sources [17]. Although the arrangement of sparse arrays can largely reduce the actual number of array elements, they often produce ambiguous angles, i.e., spurious spectral peaks, which interfere with the judgement. Taking an L-shaped uniform sparse array with an array element spacing of $2\lambda$ as an example, the spatial spectrum and its top view under the same conditions as above are shown in Figure 3.

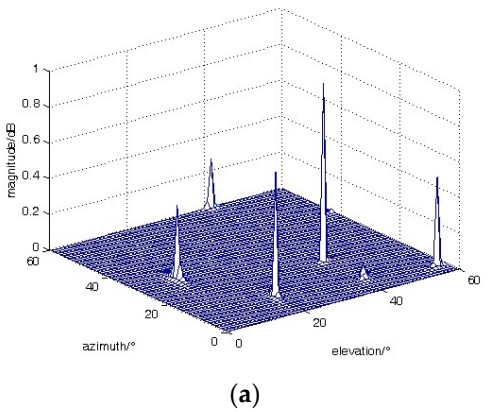

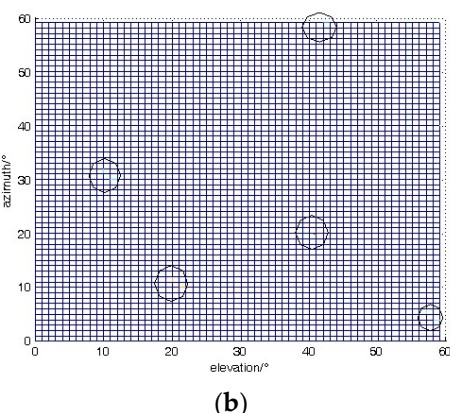

(**a**)  (**b**)

**Figure 3.** Spatial spectrum and top view of L-shaped uniform sparse array. (**a**) Spatial spectrum of L-shaped uniform sparse array. (**b**) Top view of L-shaped uniform sparse array.

Figure 3a shows that the spatial spectrum contains five distinct spectral peaks, the corresponding coordinates of which coincide with the center of the circle in Figure 3b, which are sharper than the spectral peaks in Figure 2. However, there are two blurred angles in it. To address the problem of the blurring generated by sparse arrays, the use of coprime arrays can avoid the generation of blurred angles, so coprime arrays are widely used in practice. The array element arrangements of the $X$ and $Y$ axes are changed to the mutual prime number (4, 5) and (3, 7), respectively; and the array element arrangement of the $X$ and $Y$ axis are

$$X : (0, 4, 5, 8, 10, 12, 15, 16)\lambda/2, \tag{14}$$

$$Y : (0, 3, 6, 7, 9, 12, 14, 15, 18)\lambda/2, \tag{15}$$

The number of arrays on the *X* and *Y* axis is 8 and 9, respectively; and, under the same conditions as above, the spatial spectrum and its top view were calculated as shown in Figure 4.

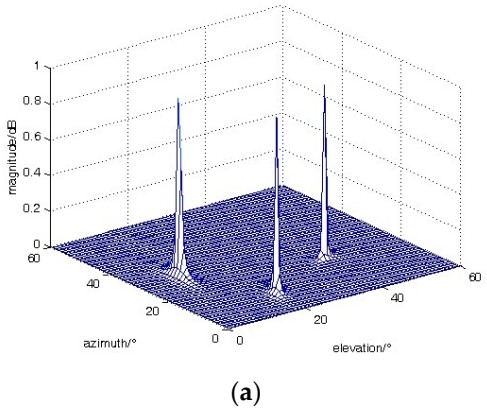

(**a**)

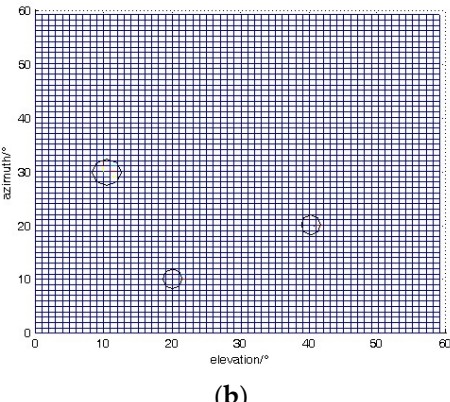

(**b**)

**Figure 4.** Spatial spectrum and top view of L-shaped coprime array. (**a**) Spatial spectrum of L-shaped coprime array. (**b**) Top view of L-shaped coprime array.

As can be seen in Figure 4, the spatial spectrum contains three spectral peaks, which correspond to the center of the circle in the top view. The sharpness of the spectral peaks is similar to that in Figure 3 and better than that in Figure 2, but there is no blurring of the angles, and the resulting elevation and azimuth angles of the target are both highly accurate. When replacing only the uniform line array in the *X* or *Y* axis with a coprime array, but not both, the spatial spectrum is obtained as follows.

The blurred spectral peaks are also avoided when the array with only one axis is replaced with a coprime array, as shown in Figure 5, which is slightly less sharp compared with those in Figures 3a and 4a. A comparison of Figure 5a,b shows that the spectral peaks are narrower in elevation when the *X* axis is a coprime array and narrower in azimuth when the *Y* axis is a coprime array but still better than that in Figure 2a, overall.

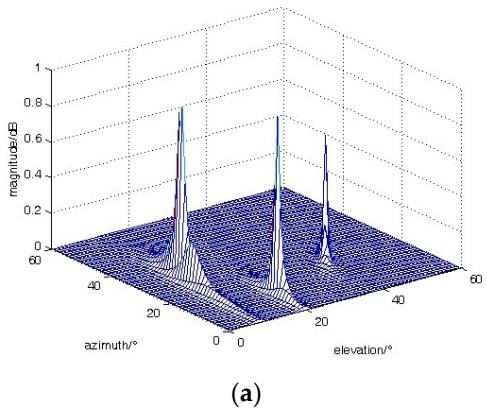

(**a**)

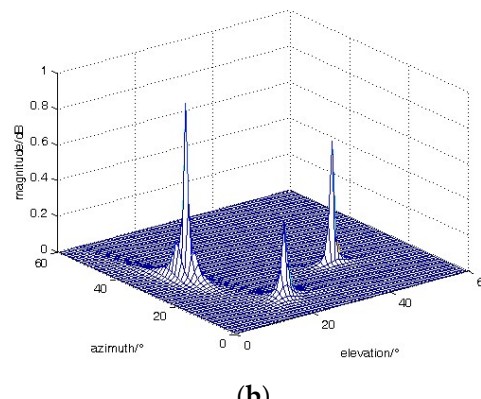

(**b**)

**Figure 5.** Spatial spectrum when the *X* or *Y* axis is a coprime array. (**a**) Coprime array in *X* axis. (**b**) Coprime array in *Y* axis.

### 2.2. Low-Elevation-Target Signal Model

The multipath effect occurs when the elevation angle of the incident to the array is low, producing a reflected wave that is coherent with the direct wave [18] in its elevation angle dimension. The multipath effect is shown in Figure 6.

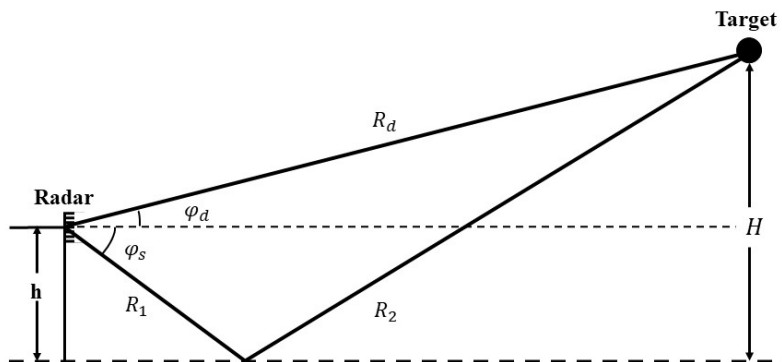

**Figure 6.** Schematic representation of the multipath effect in the elevation angle dimension.

To simplify the model, only multipath effects at the receiver side are considered. The target direct wave signal is incident on the array at an elevation angle $\varphi_d$, reflected by a smooth surface, and the reflected wave signal is incident on the array at an angle $\varphi_s(\varphi_s < 0)$. Let the height of the target be $H$ and the height of the center of the array be $h$. The difference in the wave range between the direct and reflected waves [19] is approximated as

$$\Delta R = R_d - R_s = R_d - (R_1 + R_2) \approx 2hsin\varphi_d, \tag{16}$$

According to the position relationship in Figure 6, the relationship between the direct and reflected angles satisfies Equation (17),

$$\varphi_s = \arctan(\frac{H+h}{H-h}tan\varphi_d), \tag{17}$$

When there are $K$ $(K < M + N - 1)$ incoherent sources in a space that includes $Q(Q \leq K)$ low-elevation targets, the number of signals received by the array is $(K + Q)$, which includes $Q$ direct wave signals from low-elevation targets, $Q$ reflected wave signals from low-elevation targets, and $(K - Q)$ non-low-elevation signals. The direction vectors $\boldsymbol{a}(\alpha_k)$ and $\boldsymbol{a}(\beta_k)$ for non-low-elevation targets have the same Equation (5) to (7), while the direction vectors $\boldsymbol{a}(\alpha_q)$ and $\boldsymbol{a}(\beta_q)$ for low-elevation targets can be synthesized to include both direct and reflected angles and are expressed as follows

$$\boldsymbol{a}(\alpha_q) = \boldsymbol{a}\left(\varphi_{qd}\right) + \rho\boldsymbol{a}\left(\varphi_{qs}\right), \tag{18}$$

$$X: \begin{cases} a\left(\varphi_{qd}\right) = \left[1, e^{-\frac{j2\pi dsin\varphi_{qd}cos\theta_{qd}}{\lambda}}, \ldots, e^{-\frac{j2\pi(M-1)dsin\varphi_{qd}cos\theta_{qd}}{\lambda}}\right]^T \\ a\left(\varphi_{qs}\right) = \left[1, e^{-\frac{j2\pi dsin\varphi_{qs}cos\theta_{qs}}{\lambda}}, \ldots, e^{-\frac{j2\pi(M-1)dsin\varphi_{qs}cos\theta_{qs}}{\lambda}}\right]^T \end{cases} \tag{19}$$

$$\boldsymbol{a}(\beta_q) = \boldsymbol{a}\left(\theta_{qd}\right) + \rho\boldsymbol{a}\left(\theta_{qs}\right) \tag{20}$$

$$Y: \begin{cases} a\left(\theta_{qd}\right) = \left[e^{-\frac{j2\pi dsin\varphi_{qd}sin\theta_{qd}}{\lambda}}, e^{-\frac{j2\pi 2dsin\varphi_{qd}sin\theta_{qd}}{\lambda}} \ldots, e^{-\frac{j2\pi(N-1)dsin\varphi_{qd}sin\theta_{qd}}{\lambda}}\right]^T \\ a\left(\theta_{qs}\right) = \left[e^{-\frac{j2\pi dsin\varphi_{qs}sin\theta_{qs}}{\lambda}}, e^{-\frac{j2\pi 2dsin\varphi_{qs}sin\theta_{qs}}{\lambda}} \ldots, e^{-\frac{j2\pi(N-1)dsin\varphi_{qs}sin\theta_{qs}}{\lambda}}\right]^T \end{cases} \tag{21}$$

where $\boldsymbol{a}\left(\varphi_{qd}\right)$ and $\boldsymbol{a}(\varphi_{qs})$ denote the direction vectors of the direct and reflected angles in the $X$ axis, respectively; $\boldsymbol{a}\left(\theta_{qd}\right)$ and $\boldsymbol{a}(\theta_{qs})$ denote the direction vectors of the direct and reflected angles in the $Y$ axis,; $\rho = \rho_0\exp(-j2\pi\Delta R/\lambda)$ denotes the multipath attenuation

coefficient; and $\rho_0$ denotes the specular reflection coefficient. In the spatial model, the azimuthal angles of the direct and reflected waves are equal, i.e.,

$$\theta_q = \theta_{qd} = \theta_{qs} \tag{22}$$

Substituting Equations (17) and (22) into $\boldsymbol{a}\left(\varphi_{qd}\right)$ and $\boldsymbol{a}(\varphi_{qs})$ as well as $\boldsymbol{a}\left(\theta_{qd}\right)$ and $\boldsymbol{a}(\theta_{qs})$, i.e.,

$$X: \begin{cases} \boldsymbol{a}\left(\varphi_{qd}\right) = \left(\exp\left(-j2\pi(i)dsin\varphi_{qd}cos\theta_q/\lambda\right)\right)_{1\times M}, i = 0,1, M-1 \\ \boldsymbol{a}(\varphi_{qs}) = \left(\exp\left(-j2\pi(i)dsin\left(\arctan\left(\frac{H+h}{H-h}tan\varphi_{qd}\right)\right)cos\theta_q/\lambda\right)\right)_{1\times M}, i = 0,1, M-1 \end{cases} \tag{23}$$

$$Y: \begin{cases} \boldsymbol{a}\left(\theta_{qd}\right) = \left(\exp\left(-j2\pi idsin\varphi_{qd}sin\theta_q/\lambda\right)\right)_{1\times(N-1)}, i = 1,2,\ldots,N-1 \\ \boldsymbol{a}(\theta_{qs}) = \left(\exp\left(-j2\pi(i)dsin\left(\arctan\left(\frac{H+h}{H-h}tan\varphi_{qd}\right)\right)sin\theta_q/\lambda\right)\right)_{1\times(N-1)}, i = 1,2,\ldots,N-1 \end{cases} \tag{24}$$

The direction vectors in the $X$ and $Y$ axis are

$$A_1 = \left[\boldsymbol{a}(\varphi_{1d}) + \rho\boldsymbol{a}(\varphi_{1s}),\ldots,\boldsymbol{a}\left(\varphi_{Qd}\right) + \rho\boldsymbol{a}\left(\varphi_{Qs}\right),\boldsymbol{a}\left(\alpha_{Q+1}\right),\ldots,\boldsymbol{a}(\alpha_K)\right], \tag{25}$$

$$A_2 = \left[\boldsymbol{a}(\theta_{1d}) + \rho\boldsymbol{a}(\theta_{1s}),\ldots,\boldsymbol{a}\left(\theta_{Qd}\right) + \rho\boldsymbol{a}\left(\theta_{Qs}\right),\boldsymbol{a}\left(\beta_{Q+1}\right),\ldots,\boldsymbol{a}(\beta_K)\right] \tag{26}$$

The received signal and its covariance can be calculated according to Equations (9) and (10). When the array is an L-shaped sparse array, the spacing of the array elements in the signal direction vector will change, corresponding to the sparse array element spacing, and the received signal and signal covariance matrix will change accordingly. The 2D DOA estimation becomes more complex when there is a low-elevation signal in the received signal, and the existing algorithms, whether for L-shaped uniform arrays or L-shaped sparse arrays, are not easy and accurate to implement 2D DOA estimation, and most of them can only be used for a specific array structure or a single low-elevation signal [20]. In contrast, from the signal model, there is a correspondence between the array signal covariance matrix and the elevation and azimuth angles of the targets (including low-elevation targets), i.e., in the absence of low-elevation targets, the 2D DOA relationship between the received signal covariance matrix and the target can be regarded as

$$R_x \to f((\varphi_1,\theta_1),(\varphi_2,\theta_2),\ldots,(\varphi_K,\theta_K)) \tag{27}$$

When low-elevation targets are present,

$$R_x \to f\left((\varphi_{1d},\theta_1),(\varphi_{1s},\theta_1),\ldots,(\varphi_{Qd},\theta_Q),(\varphi_{Qs},\theta_Q),(\varphi_{Q+1},\theta_{Q+1}),(\varphi_K,\theta_K)\right) \tag{28}$$

which includes $Q$ low-elevation targets, combined with Equation (17) above, Equation (23) can be further rewritten as

$$R_x \to f'\left((\varphi_{1d},\theta_1),\ldots,(\varphi_{Qd},\theta_Q),,(\varphi_{Q+1},\theta_{Q+1}),(\varphi_K,\theta_K)\right) \tag{29}$$

On this basis, the above mapping relations can be obtained with the help of deep learning, providing new ideas and methods to solve the problem of 2D DOA estimation for L-shaped uniform arrays or sparse arrays in the presence of low-elevation-angle signals.

## 3. Dilated Convolution Network Model

Due to the significant difference in the direction vector generation process between low-elevation signals and non-low-elevation signals, when there are low-elevation targets in space, conventional algorithms will first decoherence and then implement DOA estimation. The flow of the algorithm proposed in this paper is shown in Figure 7 below.

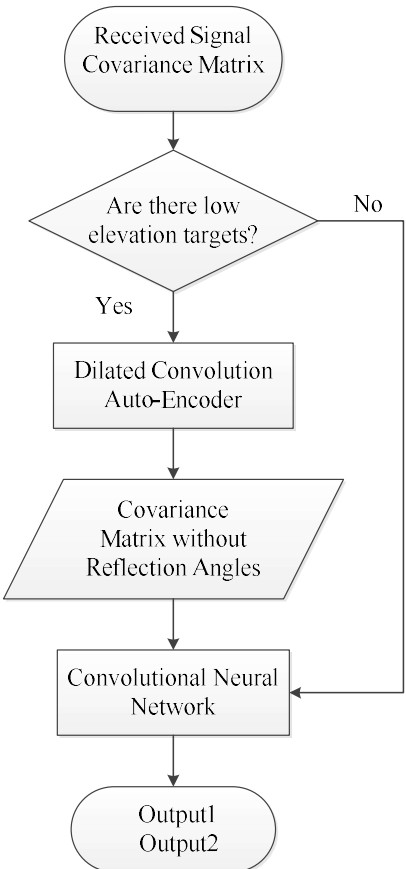

**Figure 7.** The flow of dilated convolution networks model.

As can be seen in Figure 7, when solving the DOA estimation problem, it firstly determines whether there are low-elevation targets. When there are low-elevation targets, the reflected wave components are filtered out by the dilated convolutional autoencoder (DCAE) to achieve multipath suppression, and then the 2D DOA estimation is achieved by the dilated convolutional neural network (DCNN). When there is no low-elevation target signal in space, no multipath suppression is required, so the 2D DOA estimation can be directly achieved by the DCNN. Output1 and Output2 in Figure 7 are two output branches, which are the elevation angle sequence and azimuth angle sequence, respectively, corresponding to the same position in two sequences that belongs to the same target, which can be automatically matched.

It should be added that when the covariance matrix of the received signal is used as the input to a neural network model for model training (decoherence or angle estimation), the real part of the covariance matrix is usually retained or the real and imaginary parts are stitched together to form an $N \times 2N$ ($N$ denotes the total number of array elements) real matrix, which may make the data information incomplete or affect the extraction of spatial features. In the model design process, the real and imaginary parts of the covariance matrix are expanded into a 3D matrix to form an $N \times N \times 2$ 3D matrix as the input and for training, so that the spatial features can be more fully and comprehensively extracted.

### 3.1. Dilated Convolutional Autoencoder Mode

The convolutional autoencoder is a type of autoencoder, which is a self-supervised learning algorithm that encodes and decodes data through convolutional operations so that the output data can reproduce the input data, and has a wide range of applications in data compression, data denoising, and anomaly detection [21,22]. The traditional convolutional autoencoder consists of an encoding process and a decoding process, in which the former

consists of alternating convolutional and pooling layers, with the convolutional layer used to extract features and the pooling layer used to reduce the dimensionality of the data; the latter consists of alternating deconvolution and upsampling layers, with the deconvolution being essentially the same as the convolutional layer [23], and the upsampling layer mainly achieving the recovery of data dimensionality. However, for the array received signal covariance matrix, the number of array elements is limited and the size of the covariance matrix is limited and often not very large, so the pooling layer is likely to cause insufficient feature extraction and loss of relevant features. Therefore, we discarded the pooling layer on the basis of the traditional convolutional autoencoder, and we introduced the dilation convolution to achieve data compression without data loss, DCAE model is as shown in Figure 8.

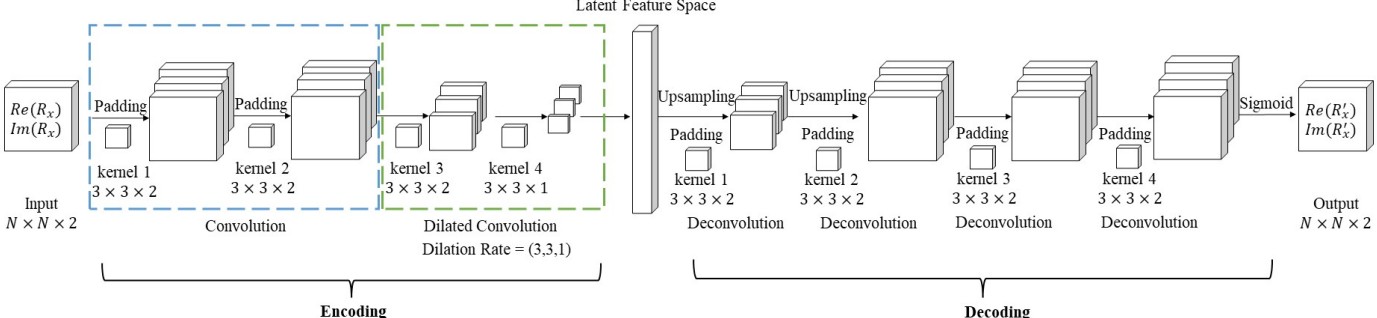

**Figure 8.** The model of DCAE.

In Figure 8, $R_x$ denotes the received signal covariance matrix containing the reflected angle, i.e., the original signal covariance matrix; $R'_x$ denotes the received signal covariance matrix without the reflected angle, containing only the direct and azimuth angles of the low-elevation target and the elevation and azimuth angles of the non-low-elevation target; and $Re(*)$ and $Im(*)$ denote the real and imaginary parts of the signal, respectively. The encoding and decoding processes are abstracted into the following mapping relationships, respectively,

$$Encoding : \boldsymbol{y} = f_e(\boldsymbol{R_x}), \tag{30}$$

$$Decoding : \boldsymbol{R}'_x = f_d(\boldsymbol{y}), \tag{31}$$

Then, the dilated convolutional autoencoder action proposed in this paper can be further described as

$$\boldsymbol{R}'_x = f_d(f_e(\boldsymbol{R_x})), \tag{32}$$

This means that a mapping between a covariance matrix with reflection angles and a covariance matrix without reflection angles is achieved. In the encoding process, the convolution operation proceeds as

$$h^n = f(\boldsymbol{R} * \boldsymbol{w}^n + \boldsymbol{b}^n), \tag{33}$$

where $\boldsymbol{R}$ denotes the input 3D matrix; $w$ denotes the 3D convolution kernel, whose number is $n$; $\boldsymbol{b}^n$ denotes the bias; and $f(*)$ denotes the activation function. The decoding process performs the deconvolution operation, which is essentially the same as the convolution operation. The two convolutional layers in the blue box in Figure 8 are regular convolutional operations with padding, which aims to preserve the boundary features. The two convolutional layers in the green box are dilated convolutional operations, the sizes of the convolutional kernels are $3 \times 3 \times 2$ and $3 \times 3 \times 1$, and the dilation rate is

$(3, 3, 1)$. The loss function of the DCAE model is a binary cross-entropy function, whose expression is

$$bce = -\sum_{i=1}^{N} \sum_{j=1}^{N} \left( r_{ij} \log(r'_{ij}) + (1 - r_{ij}) \log(1 - r'_{ij}) \right), \tag{34}$$

where $N$ denotes the total number of samples; and $r_{ij}$ and $r'_{ij}$ denote the predicted and true values, respectively.

Dilated Convolution

Dilated convolution is a kind of convolution idea to address the problem of information loss caused by connecting pooling layers after the standard convolution process [24]. The principle involves adding holes to the standard convolution map, using the holes to make the original convolution kernel have a larger reception field without increasing the number of parameters and operations [25]. Taking 2D convolution as an example, the dilation rate contains two values, which represent the magnitude of the distance between the value in the convolution kernel in the horizontal and vertical directions and its adjacent value position; when the convolution kernel size is 3 × 3, the reception field at different dilation rates is as shown in Figure 9.

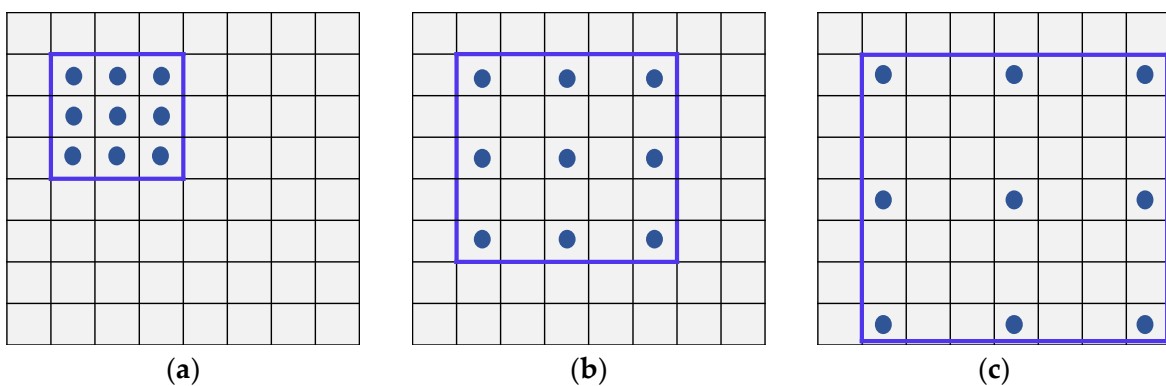

**Figure 9.** Reception fields at different dilation rates: (**a**) dilation rate = (1,1), (**b**) dilation rate = (2,2), and (**c**) dilation rate = (3,3).

The blue dots in Figure 9 represent the values of the convolution kernel, and the blue boxes represent the receptive fields under the convolution kernel; the positions in the receptive field area not filled with dots are hole., When the convolution operation is performed, the empty positions are filled with a value of 0. As shown in Figure 9a, the dilation rate is (1, 1), which is the standard convolution; the values in the convolution kernel are adjacent to each other; and the sizes of the receptive field and the convolution kernel are the same. The dilation rate in Figure 9b is (2, 2), i.e., the difference in the position between adjacent values in the convolution kernel is 2, so when the dilation rate is (2, 2), the size of the receptive field is the same as when the convolution kernel is 5 × 5. In Figure 9c, the dilation rate is (3, 3), the difference in the position of the values is 3, and the size of the receptive field is 7 × 7. The dilated convolution achieves an increase in the receptive field with the same convolution kernel and avoids an increase in computational effort.

### 3.2. Dilated Convolutional Neural Network Model

When there is no low-elevation target in the space target or the signal covariance matrix containing the low-elevation target has been suppressed by the DCAE model, the elevation and azimuth angles of the signal are obtained by the DCNN model. The structure of the model is shown in Figure 10.

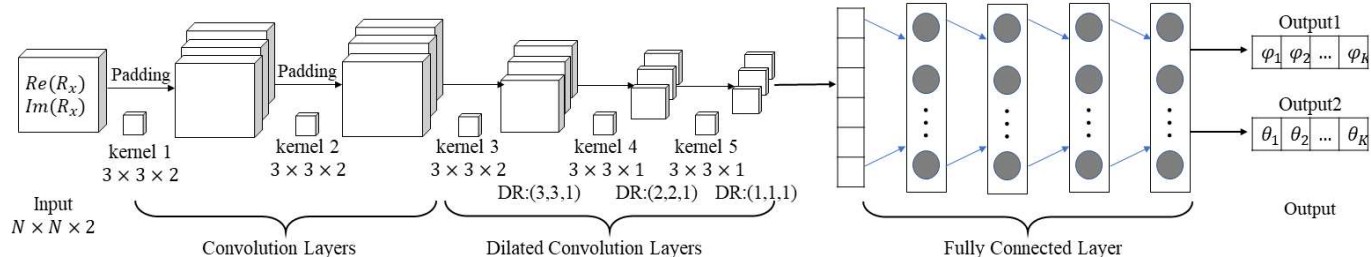

**Figure 10.** The model of DCNN. DR = dilation rate.

As can be seen in Figure 10, the model consists of five convolutional layers and four fully connected layers. The first two of the convolutional layers are standard convolutional layers with convolutional kernel sizes of $3 \times 3 \times 2$, and the last three are dilated convolutional layers with convolutional kernel sizes of $3 \times 3 \times 2$, $3 \times 3 \times 1$, and $3 \times 3 \times 1$, separately. The model contains two separate output branches for elevation and azimuth angles. Because the output of a convolutional neural network is sensitive to order, the two output branches for the elevation and azimuth angles correspond in order and no matching is required. Similar to DCAE, the input to the model, which consists of the real and imaginary parts of the covariance matrix in three dimensions, and the convolutional layers in the model are all 3D convolutional operations; the pooling layer is also removed from the convolutional neural network. The activation function for both the convolutional and fully connected layers is the ReLU function [26], which is characterized by fast convergence and no saturation of gradients, so is widely used in the training of convolutional neural network models [27]. Both output branches of the model are estimated angular values, which are regression problems, so the loss function of the model is the mean squared loss function, i.e.,

$$mse = \frac{\sum_i^N \left( (\varphi_i - \varphi_i')^2 + (\theta_i - \theta_i')^2 \right)}{2N}, \tag{35}$$

where $\varphi_i$ and $\varphi_i'$ denote the real and estimated values of the elevation angle, respectively; $\theta_i$ and $\theta_i'$ denote the real and estimated values of the azimuth angle, respectively; and $N$ denotes the number of targets.

## 4. Simulation Experiments and Analysis of Results

In the simulation experiment, we used 16 array elements; 8 and 9 uniform arrays in the $X$ and $Y$ axis, respectively; and the sparse arrays were two coprime arrays with coprime numbers (4, 5) and (3, 7). The total number of arrays was 16 due to the existence of a common element at the origin. The range of low elevation angles in the spatial signal where multipath effects occur was $(0°, 10°]$, the range of non-low elevation angles was $(10°, 60°]$, and the range of azimuth angles was $[-90°, 90°]$. The DCAE and DCNN models are shown in Figures 8 and 10 above. The number of convolutional kernels for each layer of the encoding process in the DCAE model was 200, 200, 150, and 150 in order in the decoding process, i.e., 150, 150, 200, and 200. The size of the kernels and the dilation rate were set as in Figures 8 and 10. The number of neurons in each layer of the fully connected layer was 1500, 1500, 1000, and 1000 in that order. The capacity of the training set for different formations was 50,000, the size of the test set was 2000, the number of iterations was 5000, and the batch size was 100.

### 4.1. Verification of Dilated Convolutional Autoencoder Mode Validity

Test case 1: The formation is an L-shaped uniform array. There are two targets in space, one of which is a low-elevation target with direct and reflected angles of $3.216°$ and $-5.957°$ for elevation, respectively, and $38.472°$ for the azimuth; the other is a non-low-elevation target with a $38.293°$ elevation and $48.506°$ azimuth; SNR is 10 dB; and snapshot is 100. After multipath suppression, the angle was estimated by the 2D MUSIC algorithm (the 2D search angle interval was $1°$), and the spatial spectrum is shown in Figures 11 and 12.

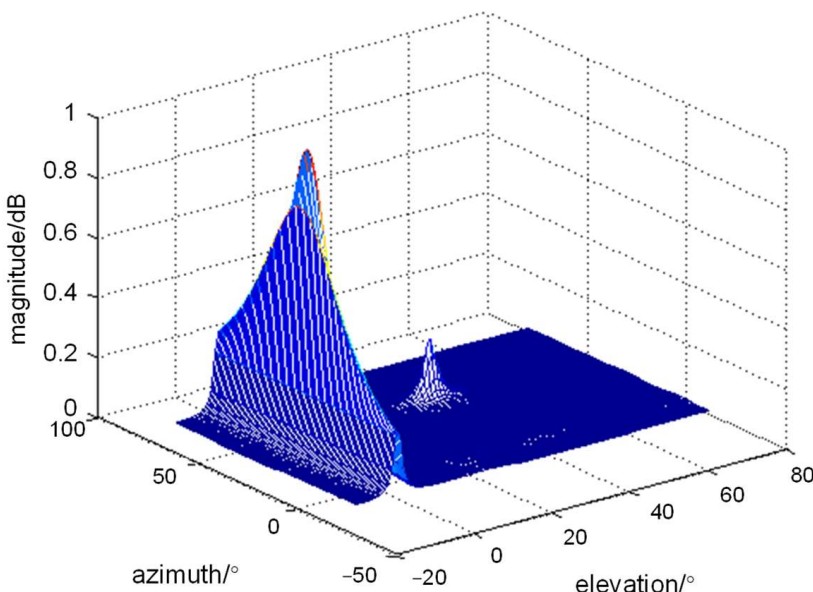

**Figure 11.** Spatial spectrum of test case 1 obtained by DCAE MUSIC.

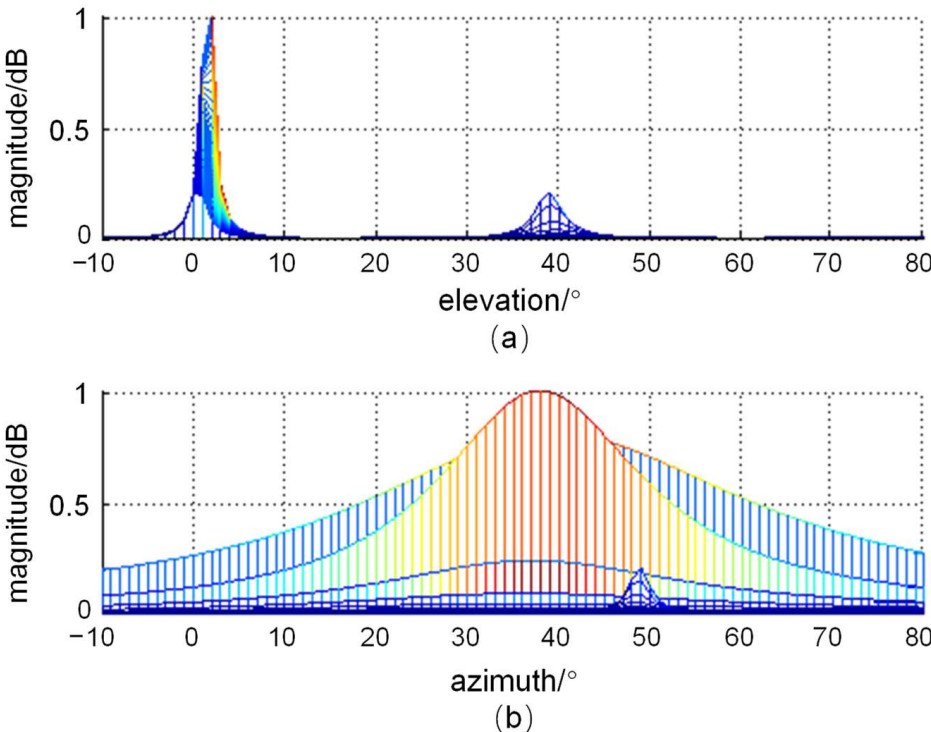

**Figure 12.** Spatial spectrum of (**a**) elevation and (**b**) azimuth angles of test case 1 obtained by DCAE MUSIC.

As can be seen in Figures 11 and 12, there are two distinct spectral peaks in the spatial spectrum, corresponding to the low-elevation target and the non-low-elevation target in the signal; there is no spectral peak for the reflected angle in Figure 12a. Comparing the azimuth of the low-altitude target, the spectral peaks of its elevation angle are sharper, while the difference between the sharpness of the azimuth and elevation angles of the non-low-altitude target is not significant. From the angle estimation accuracy and Figure 12a,b, we concluded that the elevation and azimuth angles of the low-altitude target are about 3° and 38°, respectively; and the elevation and azimuth angles of the non-low-altitude target are about 38° and 49°, respectively, which are close to the target angle in test case 1, for

the existence of a low-elevation target in space. The DCAE model can effectively filter the reflected angular component of a low-elevation target and a non-low-elevation target in space, without interfering with the direct angle of arrival and the non-low-elevation target.

Test case 2: The array is L-shaped sparse array. There are two low-elevation targets and one non-low-elevation target in space, where the direct and reflected angles of the low-elevation targets are 1.999° and −3.708°, and 8.000° and −18.937°, respectively; the azimuth angles of the two low-elevation targets are 1.904° and 1.478°; the elevation and azimuth angles of the non-low-elevation targets are 33.743° and 30.509°, respectively; SNR is 10 dB; and the snapshot is 100. After filtering the reflected angle, the angle was estimated by the 2D MUSIC algorithm (the 2D search angle interval was 1°), and the spatial spectrum is shown in Figure 13.

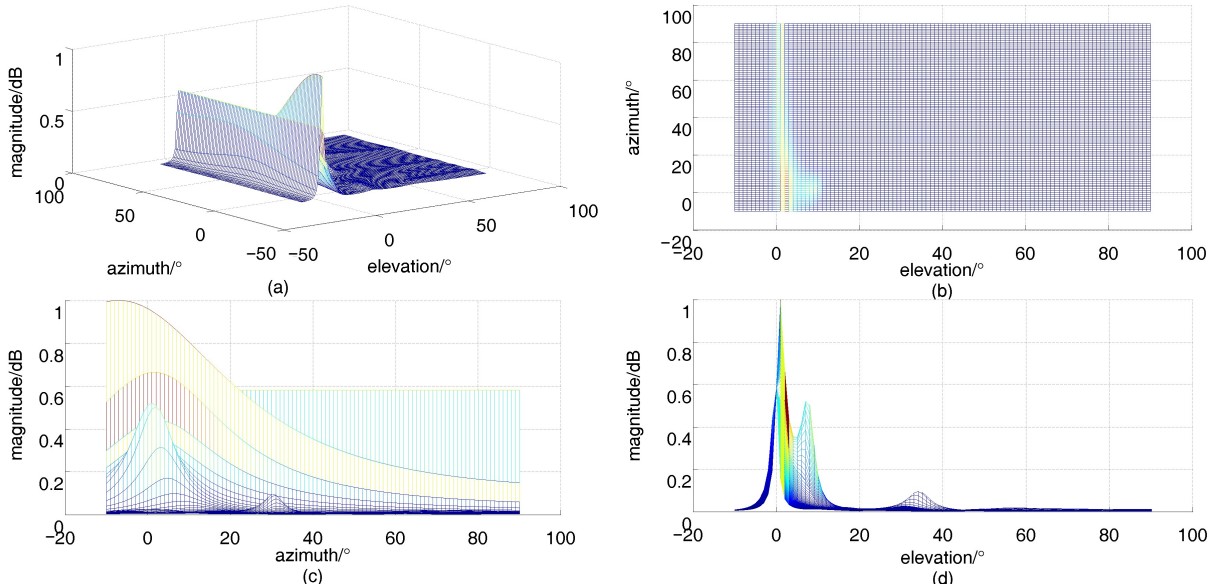

**Figure 13.** Spatial spectrum of test case 2 obtained by DCAE MUSIC: (**a**) 3D view of the 2D spatial spectrum, (**b**) top view of the 2D spatial spectrum, (**c**) spatial spectrum of azimuth angles, and (**d**) spatial spectrum of elevation angles.

Figure 13a shows the spatial spectrum of the 2D search, and Figure 13b shows the top view of the spatial spectrum, from which it can be seen that there are three spectral peaks in the spatial spectrum, with the non-low-elevation target having the lowest peak value. Figure 13c shows the azimuth angle, containing three spectral peaks corresponding to 1°, 2°, and 31°; Figure 12d shows the elevation angle, also containing three spectral peaks corresponding to 1°, 8° and 33°. The elevation and azimuth angles of the three targets obtained from Figure 13 are essentially the same as the actual angles in test case 2 and are not affected by the formation. When the three targets in test case 2 were estimated with 2D MUSIC (without the reflected angles), the spatial spectrum was obtained as shown in Figure 14.

Comparing Figures 13 and 14, the two spatial spectral distributions are basically the same. We verified that when there are multiple low-elevation targets in space, the DCAE algorithm can effectively suppress multipath without interfering with the estimation of direct-angle and non-low-elevation targets. Test cases 1 and 2 verify that the DCAE model can effectively achieve "de-multipathing" and that the DCAE model is valid.

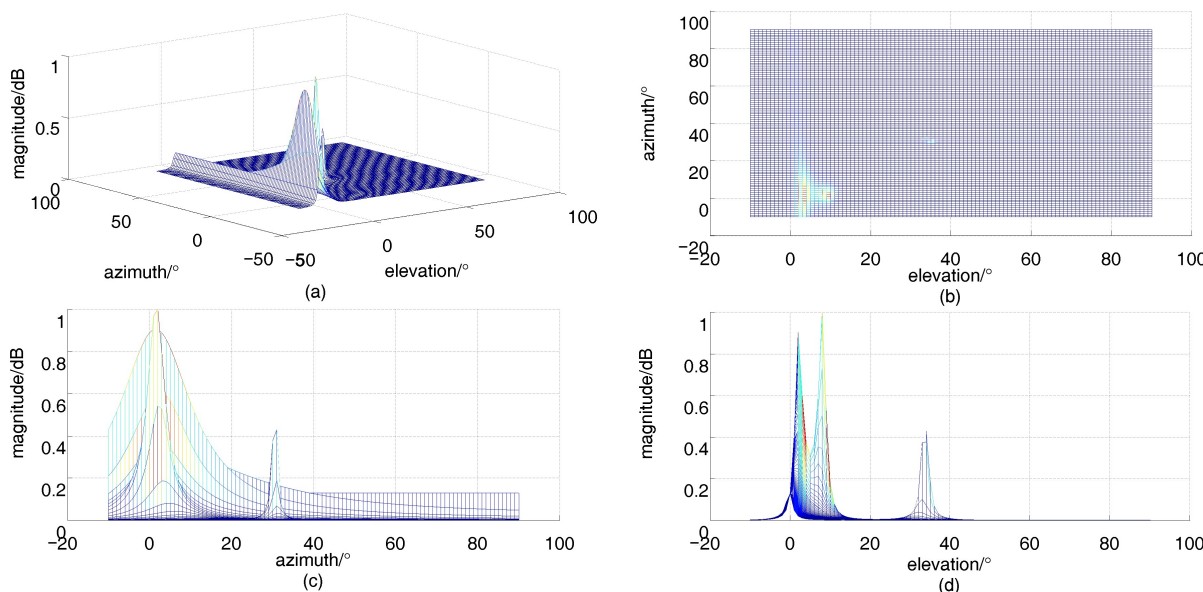

**Figure 14.** Spatial spectrum of test case 2 (without the reflected angles) obtained with 2D MUSIC: (**a**) 3D view of the 2D spatial spectrum. (**b**) Top view of the 2D spatial spectrum. (**c**) Spatial spectrum of azimuth angles. (**d**) Spatial spectrum of elevation angles.

### 4.2. Verification of Dilated Convolutional Neural Network Model Validity

The arrays were L-shaped uniform (LUA) and L-shaped sparse array (LSA), as described above; SNR was 10 dB; snapshot was 100; the numbers of targets were 2 and 3, respectively; and all were non-low-elevation targets. The efficiency rate, matching rate, and root mean square error of the angle estimation were used as the performance evaluation metrics of DCNN for 2D DOA estimation. When the angular error in the output is not greater than 5°, the angle is regarded as a valid angle. The proportion of valid angles to all output angles is the efficiency rate $P_E$; the matching rate $P_M$ indicates the proportion of azimuth and elevation angles that are accurately matched according to their positions, and the root mean square error (RMSE) is a common measure in DOA angle estimation; its expression is

$$RMSE = \sqrt{\frac{1}{N}\sum_{i=1}^{N}(\theta'_i - \theta_i)^2},\qquad(36)$$

where $N$ denotes the total number of test sets, $\theta'_i$ denotes the angle estimate output by the model, and $\theta_i$ denotes the actual angle value. After 200 Monte Carlo experiments, $P_E$, $P_M$, and RMSE were statistically obtained as shown in Table 1,

**Table 1.** $P_E$, $P_M$, and RMSE for non-low-elevation targets obtained with DCNN.

| Type | Target Number | $P_E$/% | $P_M$/% | RMSE/° | RMSE$_e$/° | RMSE$_a$/° |
|------|---------------|---------|---------|--------|-----------|-----------|
| LUA  | 2             | 99.98   | 100     | 0.3697 | 0.2901    | 0.4352    |
|      | 3             | 99.97   | 100     | 0.3083 | 0.2710    | 0.3412    |
| LSA  | 2             | 99.99   | 100     | 0.3507 | 0.2548    | 0.4256    |
|      | 3             | 100     | 100     | 0.2768 | 0.2602    | 0.2759    |

In Table 1, RMSE/° indicates the RMSE for all outputs; RMSE$_e$ and RMSE$_a$ denote the RMSE for elevation and azimuth angles respectively. From Table 1, it can be seen that all $P_E$ values are close to 100%, $P_M$ reaches 100%, and the elevation and azimuth angles of the targets in the two output branches can achieve one-to-one correspondence without parameter matching. From the RMSE results, 2D DOA estimation accuracy is better than that of the L-shaped uniform array when the array type is L-shaped sparse array, and the

estimation effect is better than that when the number of sources is three than when the number of sources is two. The estimation accuracy for elevation angle is slightly higher than that for azimuth angle in all conditions above.

When there were three targets in a space that contains two low-elevation-angle targets, $P_E$, $P_M$, and RMSE were calculated after 200 Monte Carlo experiments using the DCAE-DCNN and DCNN models (without the reflected angle in the model output); the results are shown in Table 2.

**Table 2.** $P_E$, $P_M$, and RMSE for low-elevation targets obtained with DCNN and DCAE-DCNN. "Low" and "Non-Low" denote the elevation angle of low-elevation target and non-low-elevation target in all targets, respectively.

| Type | Model | $P_E$/% | $P_E$/% | RMSE/° | RMSE$_e$/° | | RMSE$_a$/° |
| --- | --- | --- | --- | --- | --- | --- | --- |
| | | | | | Low | Non-Low | |
| LUA | DCNN | 85.74 | 97.35 | 1.6782 | 1.5757 | 1.5302 | 1.7946 |
| | DCAE-DCNN | 99.98 | 100 | 0.3441 | 0.2913 | 0.2893 | 0.3898 |
| LSA | DCNN | 87.36 | 96.77 | 1.6813 | 1.6710 | 1.6276 | 1.7125 |
| | DCAE-DCNN | 99.99 | 100 | 0.2975 | 0.2851 | 0.2720 | 0.3124 |

Table 2 shows that when only DCNN was applied for the 2D DOA estimation for multiple targets including low-elevation targets, it was not effective. Despite the high $P_E$, $P_M$ is low, and the RMSEs of the elevation and azimuth angles are much higher than those of DCAE-DCNN method, which indicates that the 2D DOA estimation problem could not be directly solved when the signal contained low-elevation targets using the DCNN method alone. As such, decoherence or de-multipathing of the received signal is necessary. Additionally, when the DCAE-DCNN algorithm was used, $P_E$ and $P_M$ were close to 100%, RMSEs were lower, and $P_E$ and $P_M$ were higher when the array type was LSA than LUA. The RMSE of the elevation angle was slightly lower than that of the azimuth angle; the RMSEs of the non-low-elevation targets were slightly lower than those of the low-elevation targets. Comparing Tables 1 and 2, the results using the DCAE-DCNN algorithm when low-elevation angle targets are present in the signal are similar to those when only the DCNN algorithm is used when low-elevation angle targets are not present. The RMSE of the former is slightly higher, which proves that the DCNN algorithm is effective and stable, and the DCAE-DCNN algorithm has a better estimation effect for the presence of low-elevation targets.

*4.3. RMSE of 2D DOA Estimation at Different SNRs with Non-Low-Elevation Targets*

In general, the variation in the SNR has a significant effect on the accuracy of DOA estimation. In this set of simulation experiments, 3 non-low-altitude targets were in space, array types were LUA and LSA, the number of snapshots was 200, and SNR was −10 dB, −5 dB, 0 dB, 5 dB, 10 dB, 15 dB, or 20 dB. The proposed DCNN model was used for angle estimation, and its results were compared with those of the 2D MUSIC algorithm to calculate the RMSE, as shown in Figure 15.

As can be seen from Figure 15a,b, the RMSE of both the elevation and azimuth angles decrease as SNR increases, and the higher SNR, the higher the estimation accuracy. From Figure 15a,b, it can be seen that the estimation accuracy of LSA is higher than that of LUA for the same algorithm. For the same array type, when the SNR was less than 10 dB, the estimation accuracy of both the elevation and azimuth angles significantly improved as the SNR increased, and the estimation performance of the DCNN algorithm was significantly better than that of 2D MUSIC. When the SNR was greater than 10dB, the decreasing trend of the RMSE became slower; for LUA, the DCNN algorithm's estimation accuracy for the azimuth angles was slightly lower than that of 2D MUSIC, and for elevation, it is slightly higher than that of 2D MUSIC. For LSA, DCNN algorithm's estimation accuracy for

elevation was better than that of 2D MUSIC, while the estimation accuracy for the azimuth angles was approximately equal between the two. By comparing Figure 15a,b, it can be seen that for either array type, DCNN algorithm's estimation accuracy for elevation angles is higher than that for the azimuth angles under each SNR condition, while the difference in the estimation performance of the 2D MUSIC algorithm for elevation and azimuth angles is not significant.

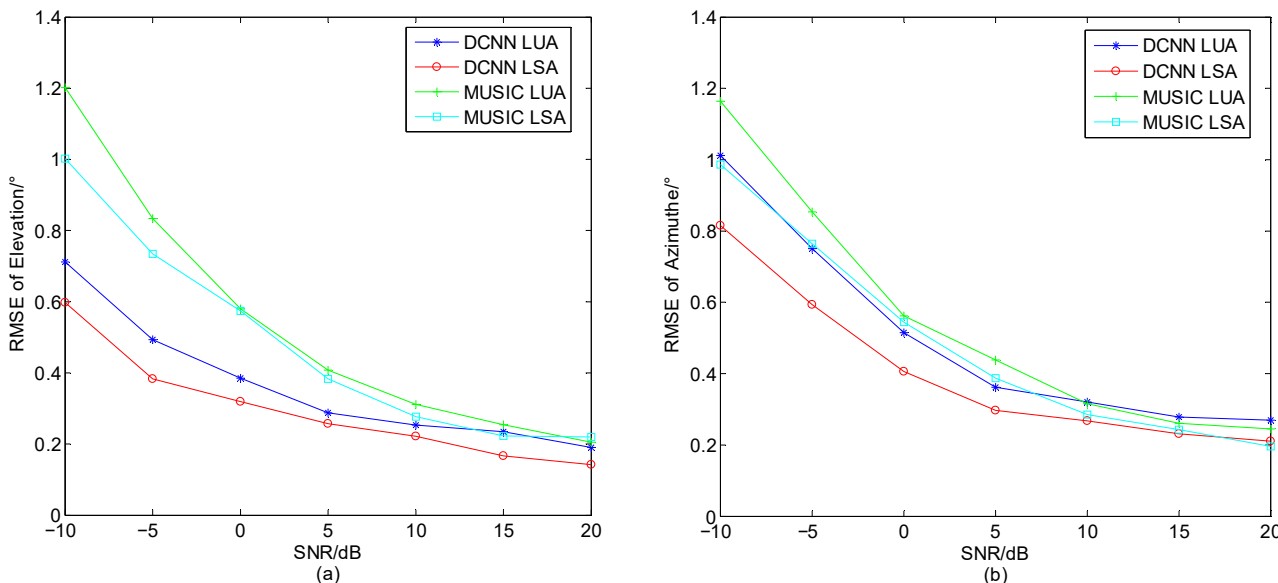

**Figure 15.** RMSE comparison at different SNRs with non-low-elevation targets: (**a**) elevation angles and (**b**) azimuth angles.

### 4.4. RMSE of 2D DOA Estimation at Different SNRs with Low-Elevation Targets

In this set of simulations, the number of spatial targets was 3, including 2 low-elevation targets; the array type was the above LUA and LSA; the number of snapshots was 200; and the SNR was −10 dB, −5 dB, 0 dB, 5 dB, 10 dB, 15 dB, or 20 dB. The DCAE-DCNN model proposed in this paper was used to perform 2D DOA estimation according to the process in Figure 7. Because most of the sparse array decoherence before 2D DOA estimation is for uniform surface arrays or other specific arrays, the Toeplitz matrix reconstruction algorithm proposed in the literature [13] and MSSP-MUSIC for 2D DOA estimation of LUA were used as comparison experiments in this set of simulations. The calculated RMSEs for the elevation and azimuth angles under each algorithm are shown in Figure 16.

In Figure 16, with the increase in the SNR, the RMSE of each algorithm for the estimation of elevation and azimuth angles shows a decreasing trend, and the estimation performance increases accordingly. When the SNR is less than 10 dB, RMSE significantly decreases as SNR increases, and the estimation accuracy of the proposed algorithm is significantly higher than that of the other two algorithms. When the SNR is greater than 10 dB, the decreasing trend in RMSE decreases, and the estimation accuracy of the proposed algorithm for LUA is close to multiple Toeplitz matrices reconstruction (MTOEP) method in the literature [13]. Comparing Figures 15 and 16, when there is a low-elevation target in the signal, the estimation accuracy of both the elevation and azimuth angles is degraded after de-multipathing by the proposed DCAE model, because when the signal is de-multipathed by the DCAE model, it may lead to new errors in the signal covariance matrix, which affects the estimation accuracy to a certain extent.

The estimation accuracy for the elevation angle is slightly higher than that for the azimuth angle for both algorithms proposed in this paper and MTOEP method proposed in the literature [13] in Figures 15 and 16. For the proposed algorithm, the reason for this is that when designing the DCNN model, the output sequence of the elevation angle is arranged in the order from smallest to largest, and the angles with the same position number in both

outputs correspond to the same target, while the order of the azimuth angle is affected by the elevation angle, resulting in the output of branch 2 being affected by branch 1. The MTOEP method in the literature [13] estimates the azimuthal angle based on the elevation angle first, so the RMSEs for the azimuthal angle of the above two algorithms are slightly larger than those for the elevation angle. However, when the MUSIC algorithm performs a two-dimensional search, it traverses the entire two-dimensional space, and the priority traversal order of the elevation and azimuthal angles does not affect the results, which are equivalent, so the difference between the RMSE of the elevation angle and azimuthal angles is not significant.

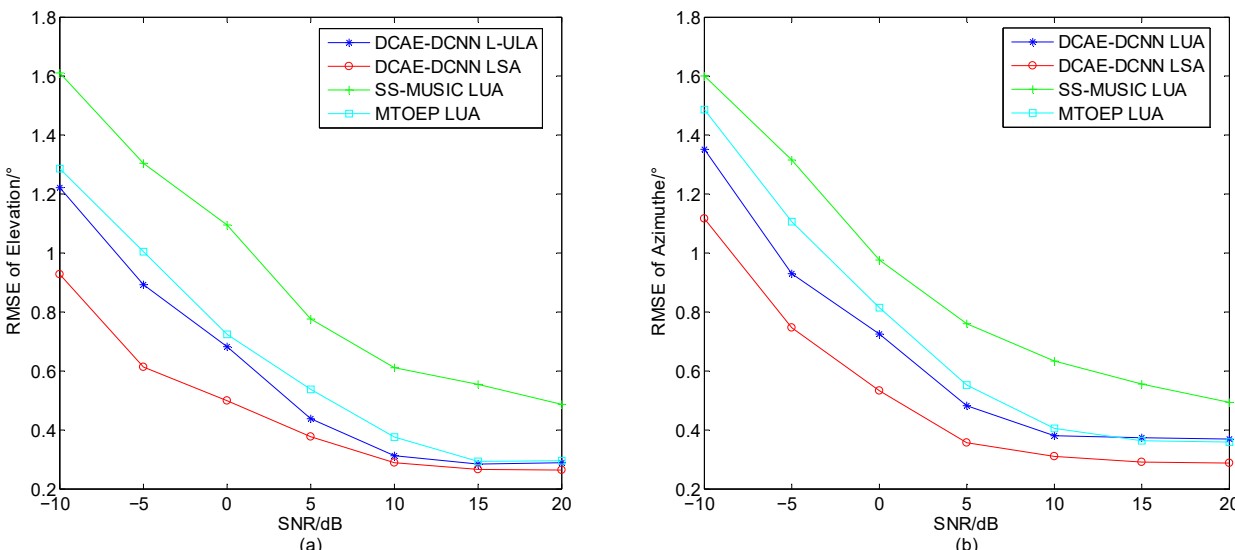

**Figure 16.** RMSE comparison at different SNRs with low-elevation targets: (**a**) elevation angles and (**b**) azimuth angles.

## 5. Discussion

For a long time, 2D DOA estimation has been of great importance in the field of array signal processing. The 2D DOA estimation of low-elevation targets, especially when the array elements are sparsely arranged, is a key and difficult problem for research in this field. The development of deep learning has provided new ideas to solve such problems. To address this problem, we developed a 2D DOA estimation algorithm based on a dilated convolutional autoencoder and s dilated convolutional neural network, which requires the total number of targets in space and the presence of low-elevation angles to be known quantities. When low-elevation targets are present in space, multipath suppression is applied to the received signal covariance matrix with DCAE, and then DCNN is used for 2D DOA estimation. Additionally, when there is no low-elevation target in space, 2D DOA estimation can be directly achieved using DCNN. The simulation experiments showed that when there are low-elevation targets in space, DCAE can effectively achieve multipath suppression and filter out the reflected angle components in the covariance matrix; when there is no low-elevation target in space or after multipath suppression is completed, DCNN can effectively achieve 2D DOA estimation with high estimation accuracy and without the need for further parameter matching.

In the proposed algorithm, the choice of hyperparameters for the model is not strict and needs to be optimized and adjusted according to the output results. In addition, we used simulation data for validation and comparison experiments, and there are certain differences between the simulation and measured data. The next study will focus on analyzing and comparing the similarities and differences between the simulation data and the measured data, so that the simulation data and the experimental scenarios can be set closer to the actual situation, thus increasing the applicability of the.

**Author Contributions:** This article was coauthored by G.H., F.Z. and B.L.; the major individual contributions are as follows: conceptualization, F.Z. and G.H.; methodology, G.H., F.Z. and B.L.; software, F.Z. and B.L.; validation, F.Z. and B.L.; formal analysis, G.H.; investigation, F.Z. and B.L.; resources, G.H.; data curation, F.Z. and B.L.; writing—original draft preparation, G.H. and F.Z.; writing—review and editing, F.Z.; supervision, G.H. and B.L.; project administration, G.H.; funding acquisition, G.H. and B.L. All authors have read and agreed to the published version of the manuscript.

**Funding:** This study was funded by the National Natural Science Foundation of China, grant number 61871395.

**Data Availability Statement:** The data presented in this study are available on request from the corresponding author. The data are not publicly available, due to the data in this paper not being from publicly available datasets but obtained from the simulation of the signal models listed in the article.

**Acknowledgments:** We thank the college for providing us with an efficient simulation platform so that we could complete the experimental simulation as scheduled. Funding from the National Natural Science Foundation of China (No. 6207011332) is gratefully acknowledged.

**Conflicts of Interest:** The authors declare no conflict of interest.

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
