# Peer review of "Estimation of the Two-Dimensional Direction of Arrival for Low-Elevation and Non-Low-Elevation Targets Based on Dilated Convolutional Networks"

_remotesensing, doi:10.3390/rs15123117_

Round 1

Reviewer 1 Report

The authors presents a 2D DOA estimation algorithm based on deep learning method. Although this method certainly has some application value, there are several points that the authors should consider revising:

1.  The title of the paper reads weird and is difficult to understand. Since abbreviation is usually not expected in the title, I would suggest the authors change the title as "Estimation on the two-dimensional direction of arrival for low elevation and non-low elevation targets based on dilated convolutional networks"

2. Many abbreviations are not defined for the first time they appear in the paper. Such as DOA in the abstract, MUSIC and ESPRIT algorithm. The readers are not necessarily familiar with these term.

3. The following paper should be cited when the author mentioned the ReLU layer in L340:

Nair, V.; Hinton, G.E. Rectified Linear Units Improve Restricted Boltzmann Machines. In Proceedings of the International Conference on Machine Learning, Haifa, Isreal, 21-24 June 2010; pp. 807-814.

4. The results of paper are only compared between DCNN and DCAE-DCNN models. How about the external models presented in previous studies, such as the MUSIC and ESPRIT algorithm mentioned in the Introduction? The authors should consider include some comparative experiment to show the advantage of the proposed method.

5. A conclusion part that summarize the main findings of the study would make the paper easier to be understood.

Reviewer 2 Report

Although the manuscript is well written, I suggest the following minor amendments to be incorporated:

Authors need to look into full manuscript for some of the below given changes that need thorough reading at least once.

1. All occurrences of et al., e.g., i.e., should be in italics.

2. Line no 441. "Table 2. This is a table. Tables should be placed in the main text near to the first time they are cited." I think authors haven't removed the line from the template.

3. Line no 469-470. Mention proper Figure No

4. Try to refine and make Figure 7 more informative. It should be the one giving an overview of methodology.

5. Line No. 211 "Then the direction vectors in the X and Y axes are" It should be axis

6. Enhance the dimensions of Figures 11-16, so that these can be clearly visible.

Reviewer 3 Report

After carefully reading the manuscript, the general comments about the strengths and weakness of this work are listed as follows:

Strengths:

(1) The paper addresses an important problem in 2D DOA estimation, specifically in the context of L-shaped uniform and sparse arrays, which are commonly used in practical applications.

(2) The proposed algorithm is based on dilated convolutional networks, which have shown promising results in various signal-processing tasks, including image and speech recognition.

(3) Using a dilated convolutional autoencoder to suppress multipath signals is an innovative approach that leverages the ability of autoencoders to learn a compressed representation of input signals.

(4) The simulation experiments provide a thorough evaluation of the proposed algorithm and demonstrate its effectiveness in both L-shaped uniform and sparse arrays, as well as in low signal-to-noise ratio conditions.

(5) The paper provides a detailed explanation of the proposed algorithm, including the signal models used, the structure of the dilated convolutional autoencoder and neural network, and the specific hyperparameters are chosen.

(6)The paper provides a comprehensive comparison of the proposed algorithm with traditional DOA estimation methods, highlighting its superiority in terms of accuracy and robustness to multipath signals.

Weaknesses:

Please respond to the following comments if you need to revise your manuscript point-by-point.

(1) The paper could benefit from a more thorough discussion of the limitations of the proposed algorithm, as well as potential future research directions.

(2) While the simulation experiments provide a good evaluation of the proposed algorithm, it would be helpful to provide more specific details on the scenarios tested and the magnitude of the improvement achieved by the proposed algorithm over traditional methods.

(3) The paper could benefit from a more detailed explanation of the choice of hyperparameters and how they were tuned.

(4) The paper could benefit from an additional discussion on the scalability of the proposed algorithm to larger arrays and more complex environments.

Some concerns have been listed below:

(1) When writing technical documents such as academic papers or reports, it is important to maintain consistency in formatting to ensure clarity and readability. In particular, when presenting vectors or matrices, it is standard practice to represent them using boldface formatting to distinguish them from other text and to make them easier to identify. Therefore, as a reviewer, I kindly request that you verify that all vectors or matrices in your document are appropriately presented in boldface to ensure that the document meets standard formatting conventions. Also, the equations are not completely expressed. For example, one right bracket is missing in the definition of A2 in Eq. (2).

(2) The authors need to consider the cases in the additive noise as impulse noise. Recently, the published papers that talked about DOA estimation considered the impact of impulse noise. The authors may check the following related works.

* Zhang, J.; Chu, P.; Liao, B. DOA Estimation in Impulsive Noise Based on FISTA Algorithm. Remote Sens. 2023, 15, 565. https://doi.org/10.3390/rs15030565

* L. Cheng, J. Chao, S. Ding and Y. Dong, "A Robust Direction of Arrival Estimation Method in Impulsive Noise," 2022 4th International Conference on Intelligent Control, Measurement and Signal Processing (ICMSP), Hangzhou, China, 2022, pp. 37-40, doi: 10.1109/ICMSP55950.2022.9859224.

* Mengya Guo, Yueping Sun, Jisheng Dai, Chunqi Chang, Robust DOA estimation for burst impulsive noise, Digital Signal Processing, Volume 114, 2021, 103059.

(3) It is important to acknowledge the limitations of any research work to provide a clear and accurate understanding of the scope and implications of the study. Therefore, as a reviewer, I recommend that you thoroughly discuss and analyze the limitations of your research work. This includes identifying potential weaknesses, constraints, or areas of uncertainty in your research methodology, data sources, or analysis techniques. Addressing these limitations not only demonstrates the rigor and transparency of your work but also helps to provide a more comprehensive and nuanced interpretation of your results.
